# IoT-Applicable Generalized Frameproof Combinatorial Designs

**Bimal Kumar Roy** [†] [ID] **and Anandarup Roy** *,[†] [ID]

Indian Statistical Institute, Kolkata 700108, West Bengal, India; bimal@isical.ac.in
* Correspondence: ananda1101@gmail.com
[†] These authors contributed equally to this work.

**Abstract:** Secret sharing schemes are widely used to protect data by breaking the secret into pieces and sharing them amongst various members of a party. In this paper, our objective is to produce a repairable ramp scheme that allows for the retrieval of a share through a collection of members in the event of its loss. Repairable Threshold Schemes (RTSs) can be used in cloud storage and General Data Protection Regulation (GDPR) protocols. Secure and energy-efficient data transfer in sensor-based IoTs is built using ramp-type schemes. Protecting personal privacy and reinforcing the security of electronic identification (eID) cards can be achieved using similar schemes. Desmedt et al. introduced the concept of frameproofness in 2021, which motivated us to further improve our construction with respect to this framework. We introduce a graph theoretic approach to the design for a well-rounded and easy presentation of the idea and clarity of our results. We also highlight the importance of secret sharing schemes for IoT applications, as they distribute the secret amongst several devices. Secret sharing schemes offer superior security in lightweight IoT compared to symmetric key encryption or AE schemes because they do not disclose the entire secret to a single device, but rather distribute it among several devices.

**Keywords:** combinatorial secret sharing; secure eID; cloud storage; SBIoT





## 1. Introduction

The Internet of Things (IoT) is a rapidly growing network of interconnected devices that communicate with each other to perform various tasks. As the number of IoT devices increases, so does the need for secure communication between them. Cryptography is an essential tool for securing IoT devices, and secret sharing schemes are one of the most promising cryptographic elements for IoT. For example, the Datachest application encrypts and stores sensitive data in commercial cloud storage systems using secret sharing [1]. The application uploads the data in encrypted form, and cryptographic keys are divided into shares. Each cloud receives one share, and this solution improves the security of users' sensitive data in the cloud. In this paper, we identify the importance of applicability of secret sharing schemes to IoT, and pay particular attention to the value our proposed distribution design may introduce through frameproofness of the underlying scheme in such applications as well as possibilities for integrating multiple or multi-level systems without complete loss of distinction of the underlying individual systems.

A secret sharing scheme is a useful tool in modern cryptography. They are distinctive in distributing a secret amongst multiple devices, ensuring that no single device has access to the entire secret. This makes secret sharing schemes ideal for IoT applications where multiple devices need to work together to perform a task. For example, in a smart home system, multiple devices such as sensors, cameras, and smart locks need to communicate with each other to provide security and convenience to the homeowner. In secret sharing-based IoT (SBIoT), each cloud server is given a share constructed using a secret sharing scheme. A collection of servers can reconstruct the secret provided that they satisfy the reconstruction criteria of the underlying scheme (instead of privately owned keys in encryption-based

schemes). Such a scheme enables processing without the need of decryption. Energy efficiency refers to the total energy consumption of an IoT network, which affects the lifetime of a network [2]. It is well-known that use of a ramp-type scheme improves the security and energy efficiency in SBIoT networks [3]. It provides better security against various types of attacks, including replay attack, modification attack, selective forwarding attack, and data leakages when a passive attacker is encountered. These benefits contribute to enhancing the overall security and performance of data transfer in SBIoT networks. Using a threshold scheme enhances personal information protection for eID cards by not storing any personal information per se in the card [4]. Instead, sensitive personal information is divided into two parts for distributed storage in the client and the eID card. This ensures safety even when eID cards are lost because none of the original information can be figured out from a single secret share. With this structure, no information whatsoever on the original can be known from only the secret share in the card.

Consider $b$ players and a positive integer $\tau \leq b$. Suppose a dealer distributes a secret to these $b$ players such that any collection of $\tau$ players can reconstruct the secret with their shares, but no smaller collection of players can do so. This is called a $(\tau, b)$-*threshold secret sharing scheme* with *threshold* $\tau$. If the dealer distributes shares to $b$ players such that any collection of $\tau_1$ players can reconstruct the secret but no collection of $\tau_2$ or less players can do so (for $\tau_2 < \tau_1 \leq b$), then it is called a $(\tau_1, \tau_2, b)$-*ramp scheme*. Thus, if $\tau_1 - \tau_2 = 1$, then it is a $(\tau_1, b)$-threshold scheme. In this paper, we shall present a repairable ramp scheme, which we call a *tensor design*.

Secret sharing schemes also play a crucial role in ensuring secure data storage within cloud environments. These schemes involve the division of data into multiple shares, which are then stored on different servers. This approach provides a safeguard against any potential compromise of a single server, thereby maintaining the security of the data. In [5], the authors present an exploration of the comparative performance of Shamir's secret sharing algorithm [6] and Rabin's IDA [7] within a private cloud framework utilizing the OpenStack cloud infrastructure. The experimental results indicated that Shamir's secret sharing algorithm outperformed Rabin's IDA in terms of generating the shares and reconstructing the data. However, Rabin's IDA exhibited a lower storage overhead when compared to Shamir's secret sharing algorithm. These findings underscore the importance of considering various factors, such as generation time, reconstruction time, and storage requirements, when selecting an appropriate secret sharing scheme for secure data storage in cloud environments.

In their 2019 work, Stinson and Kacsmar [8] demonstrated non-ideal secret sharing schemes stemming from an ideal scheme (viz. Shamir scheme) as the base scheme. They presented a distribution design which was a threshold scheme with the ability to repair lost shares with a certain probability, and secure against any adversary with lesser players than the threshold. Our work further generalizes the domain over which our distribution designs are defined, in addition to providing it with easier secret reconstruction and share repairability, and securing it in more than one context. In short, we revisit the combinatorial design and some of its key properties first.

### 1.1. Combinatorial RTS

Consider the problem of securely reconstructing the lost share of a player by that player and a subset of the other players. A combinatorial solution to this problem was proposed by Stinson and Wei [9]. These schemes are termed *combinatorial RTS*. A *repairable threshold scheme (RTS)* is a $(\tau, b)$-threshold scheme in which a subset of players can repair another player's share in the event that their share is lost or corrupted, without the participation of the dealer who set up the scheme. The repairing protocol should not compromise the (unconditional) security of the threshold scheme.

*1.2. A Drawback and an Idea of Extension*

The combinatorial model proposed so far produces shares that are in a finite field $\mathbb{F}_q^k$. Whether we can extend this notion to an integer ring is the first question. In this work, we propose a method to construct a distribution design with entries from an integer ring, thus generalizing the domain. We further show that this is a ramp scheme and consequently give a method of secret reconstruction for it, which is significantly easier in comparison to [8]. The size of the authorized coalition that can recover the secret is significantly reduced in our framework. Example 3 will demonstrate the fact.

Repairability Problem

Techniques from network reliability theory are heavily used in reliability studies of these combinatorial repairable threshold schemes in a setting where players may not be available to take part in the repair of a given player's share. Reference [8] deals with the problem of reliability of such schemes and reconstruction of secrets and repairing shares without participation of the dealer.

The scheme proposed in this paper produces a far more efficient share repairability, which is possible due to the generalized domain, and based heavily on the easier secret reconstruction mentioned beforehand.

*1.3. Frameproofness*

Moving forward with the concept of repairing shares, another similar possibility was recently explored, called framing. Instead of simply specifying the minimum size of a set of players that can access the secret, suppose the dealer defines the share distribution through some other process. Say $f : \mathbf{P} \to \{0, 1\}$ (where $\mathbf{P}$ denotes the power set of the set of all players $\mathcal{P}$) such that any *coalition* of players $\mathcal{A} \subseteq \mathcal{P}$ can access the secret if and only if $f(\mathcal{A}) = 1$ (thus, in a Shamir scheme, $f(\mathcal{A}) = 1$ if and only if $|\mathcal{A}| \geq \tau$). If $\mathcal{A} \subseteq \mathcal{P}$ maps to 1 through $f$, then $\mathcal{A}$ is called an *authorized coalition*; if it maps to 0, then $\mathcal{A}$ is an *unauthorized coalition*.

Given such an *access structure* over a secret sharing scheme, suppose a coalition $\mathcal{A}$ of players can gain information about the share of a player $P \in \mathcal{P} \setminus \mathcal{A}$ dishonestly. Then $\mathcal{A}$ can wrongly accuse $P$ of releasing information about the secret that only $\mathcal{A}$ is not authorized to access, i.e., $\mathcal{A}$ can *frame P*. Framing a player (or players) evidently undermines the security of any secret sharing scheme, as it allows a group of players to access extra information about the secret illegally. Thus, it is imperative to limit such capabilities and/or size of any such coalition when constructing a combinatorial RTS. The concept of frameproofness was examined by Desmedt et al. in their recent paper [10]. In this paper, we improve the extension scheme so that no framing is possible for any coalition of smaller size than the threshold. The question of what can be the minimum size of a coalition that can frame a player under this modification currently remains open.

## 2. Results

In this paper, we first introduce an operation, the Krönecker product of two matrices, extendable to a Krönecker product of two BIBDs. Following up with some properties of this operation, we present methods to solve two inherent problems with Krönecker products; firstly, the operation does not produce a BIBD from two BIBDs, and secondly, we resolve the issue of uniqueness that arises with the introduction of this operation. Our next theorem deals with the existence of secret reconstruction, which we prove by producing an algorithm. A probabilistic proof is given next.

An immediate consequence of our results on the new scheme is its extendability to multiple BIBDs. We discuss it briefly though a dealer's algorithm. We proceed with an example to illustrate our algorithms further. We make considerable improvements on the method of share repair described in [8] for our proposed Krönecker product-induced BIBDs.

Next, we explore the concept of frameproofness for our proposed model and improve it significantly through certain changes in the model. We also prove existence of frameproofness of the modified scheme through results based on matchings of bipartite graphs.

Finally, we note the importance of secret sharing schemes in varied IoT applications, especially for their lightweight functionality, uniquely encapsulated through the non-accessibility of the full secret to any single entity, which we strengthen by frameproofness and can expand by incorporating multiple systems by our Krönecker product.

*Organization of the Paper*

Our paper starts with a brief review of the work performed by Stinson and Wei [9] in Section 3. We then move on to describe our construction, beginning with an introduction of the Krönecker product of two BIBDs in Section 4. We describe the secret reconstruction procedure for such an object illustrated through an example in Section 5. Next, we briefly describe the method of share repair and compute the corresponding repair probabilities, much like in [8], in Section 6. We then proceed to modify this scheme to give a frameproof construction in Section 7. Furthermore, we answer the question of existence of such a modified construction in Section 8. Finally, we draw the reader's attention to the applicability of our results to secret sharing applications on the Internet of Things, especially in a secure, lightweight context, in Section 9.

### 3. Stinson and Wei's Model [9]

The classical Shamir scheme is defined over a finite field $\mathbb{F}_q$ $(q \geq b + 1)$. It involves the following:

- an *initialization phase*, in which the dealer chooses distinct, non-zero public elements $x_1, x_2, \ldots, x_b$ from $\mathbb{F}_q$, and gives value $x_i$ to player $P_i$;
- a *share distribution phase* in which the dealer chooses a secret $K = a_0 \in \mathbb{F}_q$, then secretly chooses $a_1, \ldots, a_{\tau-1} \in \mathbb{F}_q$ independently and uniformly at random, and finally computes the share $y_i = a(x_i)$ $\left(\text{where } a(x) := \sum_{j=0}^{\tau-1} a_j x^j\right)$ and gives it to player $P_i$.

The combinatorial solution proposed by Stinson and Wei [9] to the share repairability problem is based on an old technique by Benaloh and Leichter, namely, giving each player a subset of shares from an underlying threshold scheme called a base scheme (which is, say, a $(\sigma, m)$-Shamir scheme over the base field $\mathbb{F}_q$). Each player is then given a certain subset of $d$ of the $m$ shares, by use of a set system (or *design*) consisting of $b$ blocks of size $d$, defined on a set of $m$ points. This design is termed the *distribution design*:

$$\begin{pmatrix} y_{11} & y_{12} & \cdots & y_{1d} \\ y_{21} & y_{22} & \cdots & y_{2d} \\ \vdots & & & \\ y_{b1} & y_{b2} & \cdots & y_{bd} \end{pmatrix}, \qquad \left| \{y_{ij}\}_{\substack{i \in \{1,2,\ldots,b\} \\ j \in \{1,2,\ldots,d\}}} \right| \leq m. \tag{1}$$

The resulting expanded $(\tau, b)$-threshold scheme consists of each player $P_i$ corresponding to a block $B_i \in \mathcal{B}$ of the distribution design. For each point $x \in B_i$, the player $P_i$ is given the subshare $s_x$. If X denotes the set of $m$ points on which the design is defined and $\mathcal{B} = \{B_1, \ldots, B_b\}$ is the set of all blocks, then this forms an $(X, \mathcal{B})$-distribution design.

**Definition 1.** *Suppose* $2 \leq k < v$. *A* $(b, v, k, r, \lambda)$-*balanced incomplete block design or a* $(b, v, k, r, \lambda)$-*BIBD is a design* $(X, \mathcal{B})$ *such that:*

1. $|X| = v$;
2. *each block* $B \in \mathcal{B}$ *contains exactly k points;*
3. *every pair of distinct points from X is contained in exactly λ blocks.*

*Observe that if each point occurs in exactly r blocks, then the parameters $b, v, k, r, \lambda$ of a BIBD satisfy the following relations [11]:*

*(i)*      *$bk = vr$;*
*(ii)*     *$\lambda(v - 1) = r(k - 1)$;*
*(iii)*    *$b \geq v$ (and hence $r > k$).*

**Definition 2.** *We shall call a distribution design a* tensor design *if it simply satisfies property (i) above.*

*Design Properties*

Next, we consider an example to demonstrate the fact that the object constructed in Section 4 is in fact a ramp scheme. For the purpose of computations, we recall some results from [8] on block designs.

**Theorem 1** (Replication Number). *Every point in a $(v, k, \lambda)$-BIBD occurs in exactly $r = \frac{\lambda(v-1)}{k-1}$ blocks. The value r is termed the* replication number *of the scheme.*

**Theorem 2** (Blocks and Block Size). *A $(v, k, \lambda)$-BIBD has exactly $b = \frac{vr}{k} = \frac{\lambda(v^2-v)}{k^2-k}$ blocks of size k.*

## 4. Tensor Design Generated by Two BIBDs

Given two matrices $\mathcal{A}$ and $\mathcal{B}$, the usual matrix product operation can be carried out only when the column size of the left matrix $\mathcal{A}$ is equal to the row size of the right matrix $\mathcal{B}$. The Krönecker product can be applied on any two matrices, irrespective of their dimension. This operation has several applications in Linear Algebra, of which, we consider some properties that shall be useful for working with BIBDs.

### 4.1. Definition of the Krönecker Product

The *Krönecker product* of two matrices $\mathcal{A}_{b_1 \times k_1}$ and $\mathcal{B}_{b_2 \times k_2}$ is the block matrix

$$\mathcal{A} \otimes \mathcal{B} = \begin{pmatrix} \mathfrak{a}_{11}\mathcal{B} & \mathfrak{a}_{12}\mathcal{B} & \dots & \mathfrak{a}_{1k_1}\mathcal{B} \\ \mathfrak{a}_{21}\mathcal{B} & \mathfrak{a}_{22}\mathcal{B} & \dots & \mathfrak{a}_{2k_1}\mathcal{B} \\ \vdots & & & \\ \mathfrak{a}_{b_1 1}\mathcal{B} & \mathfrak{a}_{b_1 2}\mathcal{B} & \dots & \mathfrak{a}_{b_1 k_1}\mathcal{B} \end{pmatrix}, \tag{2}$$

where $\mathfrak{a}_{ij}$ denotes the entry in the *i*th row and *j*th column of $\mathcal{A}$.

Observe that Krönecker products follow the associative property. Thus, for matrices $\mathcal{A}$, $\mathcal{B}$, and $\mathcal{C}$,

$$(\mathcal{A} \otimes \mathcal{B}) \otimes \mathcal{C} = \mathcal{A} \otimes (\mathcal{B} \otimes \mathcal{C}).$$

Another interesting property of Krönecker products is that they maintain structure over block matrices. Thus, if $\mathcal{A}$ is written as a block matrix

$$\begin{pmatrix} \mathcal{A}_{11} & \mathcal{A}_{12} & \cdots & \mathcal{A}_{1k} \\ \mathcal{A}_{21} & \mathcal{A}_{22} & \cdots & \mathcal{A}_{2k} \\ \vdots & & & \\ \mathcal{A}_{b1} & \mathcal{A}_{b2} & \cdots & \mathcal{A}_{bk} \end{pmatrix} \text{ for some } b \leq b_1 \text{ and } k \leq k_1,$$

$$\text{then } \mathcal{A} \otimes \mathcal{B} = \begin{pmatrix} \mathcal{A}_{11} \otimes \mathcal{B} & \mathcal{A}_{12} \otimes \mathcal{B} & \cdots & \mathcal{A}_{1k} \otimes \mathcal{B} \\ \mathcal{A}_{21} \otimes \mathcal{B} & \mathcal{A}_{22} \otimes \mathcal{B} & \cdots & \mathcal{A}_{2k} \otimes \mathcal{B} \\ \vdots & & & \\ \mathcal{A}_{b1} \otimes \mathcal{B} & \mathcal{A}_{b2} \otimes \mathcal{B} & \cdots & \mathcal{A}_{bk} \otimes \mathcal{B} \end{pmatrix}. \tag{3}$$

## 4.2. Krönecker Product of Two BIBDs

Let $\mathcal{A}$ and $\mathcal{B}$ be the share matrices generated by ramp schemes with, respectively, $b_1$ and $b_2$ blocks having shares of sizes $k_1$ and $k_2$. Suppose $\mathcal{A}$ and $\mathcal{B}$ also denote the $b_1 \times k_1$ and $b_2 \times k_2$ matrices corresponding to the two schemes. The Krönecker product of $\mathcal{A} \otimes \mathcal{B}$ is therefore

$$M = \begin{pmatrix} \mathfrak{a}_{11}\mathcal{B} & \mathfrak{a}_{12}\mathcal{B} & \dots & \mathfrak{a}_{1k_1}\mathcal{B} \\ \mathfrak{a}_{21}\mathcal{B} & \mathfrak{a}_{22}\mathcal{B} & \dots & \mathfrak{a}_{2k_1}\mathcal{B} \\ \vdots & & & \\ \mathfrak{a}_{b_1 1}\mathcal{B} & \mathfrak{a}_{b_1 2}\mathcal{B} & \dots & \mathfrak{a}_{b_1 k_1}\mathcal{B} \end{pmatrix} = \begin{pmatrix} T_1 \\ T_2 \\ \vdots \\ T_{b_1} \end{pmatrix}, \tag{4}$$

where $T_i$ ($i \in \{1, 2, \dots, b_1\}$) is the $i$th row-block submatrix of $M$ containing rows $(i-1)b_2 + 1, (i-1)b_2 + 2, \dots, ib_2$. If the share matrix $\mathcal{A}$ is defined over the field $\mathbb{F}_{p_1}$ and $\mathcal{B}$ over the field $\mathbb{F}_{p_2}$ for some primes $p_1$ and $p_2$, then we define the scalar multiplication by simple integer multiplication:

$$\mathbb{F}_{p_1} \times \mathbb{F}_{p_2} \quad \rightarrow \quad \mathbb{Z}$$
$$\text{such that } (x_1, x_2) \quad \mapsto \quad x_1 \cdot x_2.$$

The reason behind taking such a multiplication is that the product elements are not distinguishable from integers. Therefore, $M$ is a matrix over the integer ring $\mathbb{Z}$.

At this point, the first observation that we make is that the Krönecker product $\mathcal{A} \otimes \mathcal{B}$ of two BIBDs $\mathcal{A}$ and $\mathcal{B}$ does not always produce a BIBD. To illustrate the fact, we start with a small example, and then we describe a method for resolving this issue. Also, the Krönecker product in general does not produce an injective mapping from $\mathcal{M}_{b_1 \times k_1} \times \mathcal{M}_{b_2 \times k_2}$ to the matrix space $\mathcal{M}_{b_1 b_2 \times k_1 k_2}$. So it is hopeless to search for a secret reconstruction procedure from a given Krönecker product matrix. We shall thus impose a condition producing an injective map and in turn, ensuring the existence of secret reconstruction.

Consider an example of two $2 - (4, 3, 2)$ Shamir schemes in $\mathbb{F}_5$ and $\mathbb{F}_7$ over the points $\{1, 2, 3, 4\}$ and $\{1, 2, 3, 5\}$ constructed using two polynomials modulo $\mathbb{F}_5$ and $\mathbb{F}_7$, respectively. These can be represented by share matrices $\mathcal{A}$ and $\mathcal{B}$, respectively, with $r_1 = r_2 = 3$:

$$\mathcal{A} = \begin{pmatrix} 1 & 2 & 3 \\ 2 & 1 & 4 \\ 3 & 4 & 2 \\ 4 & 3 & 1 \end{pmatrix} \text{ and } \mathcal{B} = \begin{pmatrix} 1 & 2 & 3 \\ 2 & 3 & 5 \\ 3 & 5 & 1 \\ 5 & 1 & 2 \end{pmatrix}. \tag{5}$$

The Krönecker product of the BIBDs $\mathcal{A}$ and $\mathcal{B}$ is as follows:

| 1 | 2 | 3 | 2 | 4 | 6 | 3 | 6 | 9 |
|----|----|----|----|----|----|----|----|----|
| 2 | 3 | 5 | 4 | 6 | 10 | 6 | 9 | 15 |
| 3 | 5 | 1 | 6 | 10 | 2 | 9 | 15 | 3 |
| 5 | 1 | 2 | 10 | 2 | 4 | 15 | 3 | 6 |
| 2 | 4 | 6 | 1 | 2 | 3 | 4 | 8 | 12 |
| 4 | 6 | 10 | 2 | 3 | 5 | 8 | 12 | 20 |
| 6 | 10 | 2 | 3 | 5 | 1 | 12 | 20 | 4 |
| 10 | 2 | 4 | 5 | 1 | 2 | 20 | 4 | 8 |
| 3 | 6 | 9 | 4 | 8 | 12 | 2 | 4 | 6 |
| 6 | 9 | 15 | 8 | 12 | 20 | 4 | 6 | 10 |
| 9 | 15 | 3 | 12 | 20 | 4 | 6 | 10 | 2 |
| 15 | 3 | 6 | 20 | 4 | 8 | 10 | 2 | 4 |
| 4 | 8 | 12 | 3 | 6 | 9 | 1 | 2 | 3 |
| 8 | 12 | 20 | 6 | 9 | 15 | 2 | 3 | 5 |
| 12 | 20 | 4 | 9 | 15 | 3 | 3 | 5 | 1 |
| 20 | 4 | 8 | 15 | 3 | 6 | 5 | 1 | 2 |

Hence, $\mathcal{A} \otimes \mathcal{B}$ has the parameters $b = 16, v = 12$, and $k = 9$; the parameters $r$ and $\lambda$ are not well-defined. Obviously, neither does this satisfy property 3 of a BIBD (Definition 1), nor the relation (i) of a tensor design (Definition 2). Lemmas 1–3 and Theorem 3 ensure that we always obtain a tensor design from a Krönecker product, and furthermore that we always obtain a secret reconstruction for such a share distribution scheme.

*4.3. Some Results on the Krönecker Product of BIBDs*

We now resolve these issues by defining some properties of a tensor design. Let $\mathcal{A}$ and $\mathcal{B}$ be share matrices defined on points $\{x_1, x_2, \ldots, x_n\}$ and $\{y_1, y_2, \ldots, y_m\}$, respectively. Let $\mathcal{B}_d$ be the same distribution scheme as $\mathcal{B}$, but on the points $\{y_1 + d, y_2 + d, \ldots, y_m + d\}$. The position of an element in the Krönecker product of these two matrices can be found by simple counting, and is stated in the following lemma:

**Lemma 1.** *The product of $a_{ij} \in \mathcal{A}$ and $b_{kl} \in \mathcal{B}$ can be found in the row $(i - 1)b_2 + k$ (which is also the player number in the repair scheme represented by M), and the column $(j - 1)k_2 + l$ of M.*

The next result helps ensure that $\mathcal{A} \otimes \mathcal{B}$ is indeed a BIBD:

**Lemma 2.** *Let $\{x_1, x_2, \ldots, x_n\}$ and $\{y_1, y_2, \ldots, y_m\}$ be two collections of integers. Then there exists an integer d such that $\{x_1, x_2, \ldots, x_n\}$ and $\{y_1 + d, y_2 + d, \ldots, y_m + d\}$ have no multiplicative collisions of the type $x_i y_j = x_k y_l$ for $(i, j) \neq (k, l)$.*

**Proof.** Set $d \geq \max\limits_{\substack{i,k\in\{1,2,\ldots,n\} \\ j,l\in\{1,2,\ldots,m\}}} \{x_i y_j - x_k y_l\} + 1$. Suppose $x_i(y_j + d) = x_k(y_l + d)$.

$$
\begin{aligned}
\implies x_i y_j + x_i d &= x_k y_l + x_k d \\
\implies (x_k - x_i)d &= x_i y_j - x_k y_l \\
\implies d &= \frac{x_i y_j - x_k y_l}{x_k - x_i};
\end{aligned}
\tag{6}
$$

however, since $d \geq \max\limits_{\substack{i,k\in\{1,2,\ldots,n\} \\ j,l\in\{1,2,\ldots,m\}}} \{x_i y_j - x_k y_l\} + 1$, this is a contradiction. Therefore, $\{x_1, x_2, \ldots, x_n\}$ and $\{y_1 + d, y_2 + d, \ldots, y_m + d\}$ produce no multiplicative collisions. □

**Lemma 3.** *Given a list of distinct elements $\{y_1, y_2, \ldots, y_m\}$, we can choose an integer $\hat{d}$ such that $\gcd(y_1 + \hat{d}, y_2 + \hat{d}, \ldots, y_m + \hat{d}) = 1$.*

**Proof.** Without loss of generality, we may assume $y_1 < y_2 < \cdots < y_m$. Let $l = \gcd(y_1, y_2, \ldots, y_m)$ and fix $i < j$ in $\{1, 2, \ldots, m\}$. Thus, $y_i = lk_i$ and $y_j = lk_j$ such that $k_i < k_j$. Choose $\hat{d}$ such that $\gcd(\hat{d}, l) = 1$ and $\gcd(\hat{d} + y_i, k_j - k_i) = 1$ for some j in $\{1, 2, \ldots, m\}$. Now, $\gcd(y_i + \hat{d}, y_j + \hat{d}) = \gcd(lk_i + \hat{d}, lk_j + \hat{d}) = \gcd(lk_i + \hat{d}, l(k_j - k_i)) = 1$. □

**Theorem 3** (Reconstruction from Tensor Designs). *Consider a $(v_1, k_1, \lambda_1, b_1, r_1)$-BIBD $\mathcal{A}$ and a $(v_2, k_2, \lambda_2, b_2, r_2)$-BIBD $\mathcal{B}$.*

1. *The matrix $\mathcal{A} \otimes \mathcal{B}_d$ produces a tensor design (over the integer ring $\mathbb{Z}$) for a (public) integer d such that there are no multiplicative collisions of the type $x_i(y_j + d) = x_k(y_l + d)$ for $(i, j) \neq (k, l)$.*
2. * • If $\gcd(x_1, x_2, \ldots, x_{v_1}) = 1$;*
   * • if $\gcd(y_1, y_2, \ldots, y_{v_2}) = 1$;*
   *then $\mathcal{A}$ and $\mathcal{B}$ can be reproduced from a collection of players in the new scheme $\mathcal{A} \otimes \mathcal{B}_d$, hence enabling share repair and secret reconstruction.*

This theorem can be generalized for finitely many such Krönecker products, and motivates us to present the following algorithm for a share distribution scheme.

**Proof.** The parameters of the Krönecker product $\mathcal{A} \otimes \mathcal{B}$ are $b = b_1 b_2, v = v_1 v_2$, $k = k_1 k_2, r = r + 1r + 2, \lambda = \lambda_1 \lambda_2$. Part 1 of the theorem therefore follows from Lemma 2, which ensures a well-defined value for $r$, and Lemma 3, which ensures a well-defined value for $\lambda$.

In order to prove part 2, we describe two ways to reproduce $\mathcal{A}$ and $\mathcal{B}$. Recall first that any $\tau_1$ rows of $\mathcal{A}$ produce all points of $\mathcal{A}$, and similarly $\tau_2$ rows for $\mathcal{B}_d$. Furthermore, we claim the following:

**[I]** A collection of players that has

    (i)        $\tau_2$ players from one row-block $T_i$ of $M$;

    (ii)      at least one player from distinct $\tau_1 - 1$ row-blocks $T_j \neq T_i$ of the remaining $b_1 - 1$ row-blocks

    can reconstruct the secret.

**[II]** Let $S_j$ ($j \in \{1, 2, \ldots, b_2\}$) be the collection of players $\{P_{b_2 k + j} : k \in \{0, 1, \ldots \ldots, b_1 - 1\}\}$. A collection of players that contains

    (i)        $\tau_1$ players from one $S_j$;

    (ii)      at least one player from $\tau_2 - 1$ $S_i$, $i \neq j$

    can also reconstruct the secret.

We now present an algorithm to prove claim [I]; claim [II] follows similarly.

1.    The share of the $j$th player $P_{i \cdot b_2 - 1 + j}$ of the $i$th row-block $T_i$ is of the form

$$\mathfrak{a}_{i1} \cdot \{\mathfrak{b}_{j1}, \mathfrak{b}_{j2}, \ldots, \mathfrak{b}_{jk_2}\}, \mathfrak{a}_{i2} \cdot \{\mathfrak{b}_{j1}, \mathfrak{b}_{j2}, \ldots, \mathfrak{b}_{jk_2}\}, \ldots, \mathfrak{a}_{ik_1} \cdot \{\mathfrak{b}_{j1}, \mathfrak{b}_{j2}, \ldots, \mathfrak{b}_{jk_2}\}.$$

Fix any $i \in \{1, 2, \ldots, b_1\}$ and choose $j_1, j_2, \ldots, j_{\tau_2}$ to ensure that $\gcd(\mathfrak{b}_{j_11}, \mathfrak{b}_{j_12}, \ldots, \mathfrak{b}_{j_1 k_2}, \mathfrak{b}_{j_2 1}, \mathfrak{b}_{j_2 2}, \ldots, \mathfrak{b}_{j_2 k_2}, \ldots, \mathfrak{b}_{j_{\tau_2} 1}, \mathfrak{b}_{j_{\tau_2} 2}, \ldots, \mathfrak{b}_{j_{\tau_2} k_2}) = 1.$

2.    Therefore, the values of $\mathfrak{a}_{i1}, \mathfrak{a}_{i2}, \ldots, \mathfrak{a}_{ik_1}$ become known. Divide $\mathfrak{a}_{i\alpha} \mathfrak{b}_{j_k \beta}$ by $\mathfrak{a}_{i\alpha}$ (for $\alpha \in \{1, 2, \ldots, k_1\}$, $\beta \in \{1, 2, \ldots, k_2\}$ and $k \in \{1, 2, \ldots, \tau_2\}$) to obtain $\mathfrak{b}_{j_k 1}, \mathfrak{b}_{j_k 2}, \ldots, \mathfrak{b}_{j_k k_2}$.

3.    Construct the complete matrix $\mathcal{B}_d$ using the shares of $\tau_2$ players of $\mathcal{B}_d$ that are now known. Hence construct $\mathcal{B}$.

4.    Using the values of the elements in $\mathcal{B}_d$, compute the values $\mathfrak{a}_{i'1}, \mathfrak{a}_{i'2}, \ldots, \mathfrak{a}_{i'k_1}$ for $\tau_1 - 1$ indices $i'$ that are distinct from each other as well as from $i$.

5.    Hence, construct $\mathcal{A}$ from the shares of $\tau_1$ players of $\mathcal{A}$ thus obtained.

6.    Finally compute the secret from $\mathcal{A}$ and $\mathcal{B}$.

    $\square$

This reconstruction algorithm is clearly better than the one in [8] in the sense that the size of the authorized coalition is smaller. In fact, the size of the authorized coalition, while not unique, has a lower bound in the number of players. The following section provides a proof that there is always a secret reconstruction for this scheme.

### 4.4. Proof of Existence of Secret Reconstruction

Let us redefine the problem in terms of random variables. Let $X_1, X_2, \ldots, X_n$ be sampled without replacement from the collection of all players. We assume a uniform probability distribution over the set of all players.

$$\text{Let } I_{i,j} = \begin{cases} 1 & \text{if } X_i \in S_j, i \in [n], j \in [b_2], \\ 0 & \text{otherwise.} \end{cases}$$

$$\text{Also let } J_{i,k} = \begin{cases} 1 & \text{if } X_i \in T_k, i \in [n], k \in [b_1], \\ 0 & \text{otherwise.} \end{cases}$$

We further define $n_k = \sum\limits_{i=1}^{n} J_{i,k}$ and $r_j = \sum\limits_{i=1}^{n} I_{i,j}$. Then the condition for reconstruction becomes

**[I]**

    (i)        $\max\limits_{k \in [b_1]} n_k \geq \tau_2$,

    (ii)      $n_k \geq 1$ for at least $\tau_1$ indices $k$.

**[II]**

    (i)        $\max\limits_{j \in [b_2]} r_j \geq \tau_1$,

    (ii)      $r_j \geq 1$ for at least $\tau_2$ indices $j$.

Let $E_1$ be the event that condition [I] is satisfied and $E_2$ be the event that condition [II] is satisfied. Also, let $D(n_0)$ be the event that $n \geq n_0$. We find an $n_0$ such that $\Pr[E_1 \cup E_2 \mid n \geq n_0] \approx 1$. This is equivalent to $\Pr\big[E_1^c \cap E_2^c \mid n \geq n_0\big] \approx 0$. In fact, it is sufficient to show $\Pr\big[E_1^c \mid n \geq n_0\big] \approx 0$ and $\Pr\big[E_2^c \mid n \geq n_0\big] \approx 0$.

As $E_1 = E_1(i) \cap E_1(ii)$, $E_1^c = E_1(i)^c \cup E_1(ii)^c$,

$$\Pr[E_1^c \mid n \geq n_0] = \Pr[E_1(i)^c \cup E_1(ii)^c \mid n \geq n_0]$$
$$= \Pr\Big[E_1^(i)c \mid n \geq n_0\Big] + \Pr[E_1(ii)^c \mid n \geq n_0] - \Pr[E_1(i)^c \cap E_1(ii)^c \mid n \geq n_0]$$

**Lemma 4.** $\Pr[E_1(i)^c \cap E_1(ii)^c \mid n \geq (\tau_1 - 1)(\tau_2 - 1) + 1] = 0.$

**Proof.** We observe that $E_1(i)^c$ is the event $\max\limits_{k \in [b_1]} n_k < \tau_2$ and $E_1(ii)^c$ is the event that $n_k \geq 1$ for at most $\tau_1 - 1$ indices $k$. Thus, if there are $(\tau_1 - 1)(\tau_2 - 1) + 1$ players in a collection, then by the pigeonhole principle, either $E_1(i)^c$ or $E_1(ii)^c$ is violated. $\square$

**Lemma 5.** $\Pr[E_1(i)^c \mid n \geq (\tau_2 - 1)b_1 + 1] = 0.$

**Proof.** We observe that $E_1(i)^c$ is the event $\max\limits_{k \in [b_1]} n_k < \tau_2$ and there are $b_1$ $n_k$s. Thus, if there are $(\tau_2 - 1)b_1 + 1$ players in a collection, then by the pigeonhole principle, $E_1(i)^c$ is violated, since there is at least one $n_k$ with $\tau_2$ or more players. $\square$

**Lemma 6.** $\Pr[E_1(ii)^c \mid n \geq (\tau_1 - 1)b_2 + 1] = 0.$

**Proof.** We observe that $E_1(ii)^c$ is the event that $n_k \geq 1$ for at most $\tau_1 - 1$ indices $k$. By definition, each $n_k$ can have at most $b_2$ elements. Thus, any collection of $(\tau_1 - 1)b_2 + 1$ players violates $E_1(ii)^c$. $\square$

**Lemma 7.** $\Pr[E_2(i)^c \cap E_2(ii)^c \mid n \geq (\tau_1 - 1)(\tau_2 - 1) + 1] = 0.$

**Proof.** We observe that $E_2(i)^c$ is the event $\max\limits_{j \in [b_2]} r_j < \tau_1$ and $E_2(ii)^c$ is the event that $r_j \geq 1$ for at most $\tau_2 - 1$ indices $j$. Thus, if there are $(\tau_1 - 1)(\tau_2 - 1) + 1$ players in a collection, then by the pigeonhole principle, either $E_2(i)^c$ or $E_2(ii)^c$ is violated. $\square$

**Lemma 8.** $\Pr[E_2(i)^c \mid n \geq (\tau_1 - 1)b_2 + 1] = 0.$

**Proof.** We observe that $E_2(i)^c$ is the event $\max\limits_{j \in [b_2]} r_j < \tau_1$ and there are $b_2$ $r_j$s. Thus, if there are $(\tau_1 - 1)b_2 + 1$ players in a collection, then by the pigeonhole principle, $E_2(i)^c$ is violated, since there is at least one $r_j$ with $\tau_1$ or more players. $\square$

**Lemma 9.** $\Pr[E_2(ii)^c \mid n \geq (\tau_2 - 1)b_1 + 1] = 0.$

**Proof.** We observe that $E_2(ii)^c$ is the event that $r_j \geq 1$ for at most $\tau_2 - 1$ indices $j$. By definition, each $r_j$ can have at most $b_1$ elements. Thus, any collection of $(\tau_2 - 1)b_1 + 1$ players violates $E_2(ii)^c$. □

For $n_0 = \max\{(\tau_2 - 1)b_1 + 1, (\tau_1 - 1)b_2 + 1\}$, Lemmas 4–6 imply $\Pr[E_1^c \mid n \geq n_0] = 0$ and $n_0 = \max\{(\tau_2 - 1)b_1 + 1, (\tau_1 - 1)b_2 + 1\}$, and Lemmas 7–9 imply $\Pr[E_2^c \mid n \geq n_0] = 0$.

Note that the bound given here for the reconstruction number *is tight*, as we might expect. In the example presented in Section 5, the bound turns out to be 5, which matches all the bounds above. Corresponding counterexamples can be constructed to show that no smaller-sized general collection can complete the reconstruction. This result can be generalized for three or more designs. These results provide us with the tools to present a generalized scheme, which we do now.

*4.5. A Generalized Share Distribution Scheme*

1.  Dealer selects $n$ (not necessarily distinct) BIBDs $\mathcal{A}_1, \mathcal{A}_2, \ldots, \mathcal{A}_n$, where for $i \in \{1, 2, \ldots, n\}$, $\mathcal{A}_i$ is defined over points $\{x_1^i, x_2^i, \ldots, x_{v_i}^i\}$.
2.  Dealer finds an integer $d_1$ such that $\gcd(x_1^1 + d_1, x_2^1 + d_1, \ldots, x_{v_1}^1 + d_1) = 1$.
3.  For $i \in \{2, \ldots, n\}$:
    *   Dealer finds an integer $d_i$ (using Lemmas 2 and 3) such that $d_i$ breaks all pairwise multiplicative collisions and makes the gcd of all elements $x_l^j + d_j$ ($j \in \{1, \ldots, i - 1\}$, $l \in \{1, \ldots, v_j\}$) and $x_1^i + d_i, x_2^i + d_i, \ldots, x_{v_i}^i + d_i$ is 1.
4.  $M \leftarrow \mathcal{A}_1 \otimes \mathcal{A}_2 \otimes \cdots \otimes \mathcal{A}_n$.
5.  Dealer distributes each row $i$ of $M$ as share to player $P_i$ and outputs $(d_1, d_2, \ldots, d_n)$ publicly.

Note that by Theorem 3, $M$ is a tensor design, and the algorithm in the proof of the theorem can be generalized for secret reconstruction of this scheme.

## 5. Example

Recall the previous example (5). Using the algorithm in Section 4.5, we produce a tensor design $\mathcal{A} \otimes \mathcal{B}_{21}$ using an integer $d = 21$ satisfying Lemma 3. Representing the share matrix modified from $\mathcal{B}$ by $\mathcal{B}_{21}$ (and noting that both share matrices are undeclared), with $r_1 = r_2 = 3$:

$$\mathcal{B}_{21} = \begin{pmatrix} 22 & 23 & 24 \\ 23 & 24 & 26 \\ 24 & 26 & 22 \\ 26 & 22 & 23 \end{pmatrix}, \tag{7}$$

we still have $b_1 = 4$, $b_2 = 4$, $k_1 = 3$, and $k_2 = 3$. Observe that $\tau_1 = 2$ and $\tau_2 = 2$ are the reconstruction numbers of $\mathcal{A}$ and $\mathcal{B}$, respectively. The Krönecker product of the two matrices $\mathcal{A}$ and $\mathcal{B}_{21}$, represented by the matrix $M$, is shown in Figure 1.

| 22 | 23 | 24 | 44 | 46 | 48 | 66 | 69 | 72 | |
|----|----|----|----|----|----|----|----|----|---|
| 23 | 24 | 26 | 46 | 48 | 52 | 69 | 72 | 78 | $T_1=\{P_1,P_2,P_3,P_4\}$ |
| 24 | 26 | 22 | 48 | 52 | 44 | 72 | 78 | 66 | |
| 26 | 22 | 23 | 52 | 44 | 46 | 78 | 66 | 69 | |
| 44 | 46 | 48 | 22 | 23 | 24 | 88 | 92 | 96 | |
| 46 | 48 | 52 | 23 | 24 | 26 | 92 | 96 | 104 | $T_2=\{P_5,P_6,P_7,P_8\}$ |
| 48 | 52 | 44 | 24 | 26 | 22 | 96 | 104 | 88 | |
| 52 | 44 | 46 | 26 | 22 | 23 | 104 | 88 | 92 | |
| 66 | 69 | 72 | 88 | 92 | 96 | 44 | 46 | 48 | |
| 69 | 72 | 78 | 92 | 96 | 104 | 46 | 48 | 52 | $T_3=\{P_9,P_{10},P_{11},P_{12}\}$ |
| 72 | 78 | 66 | 96 | 104 | 88 | 48 | 52 | 44 | |
| 78 | 66 | 69 | 104 | 88 | 92 | 52 | 44 | 46 | |
| 88 | 92 | 96 | 66 | 69 | 72 | 22 | 23 | 24 | |
| 92 | 96 | 104 | 69 | 72 | 78 | 23 | 24 | 26 | $T_4=\{P_{13},P_{14},P_{15},P_{16}\}$ |
| 96 | 104 | 88 | 72 | 78 | 66 | 24 | 26 | 22 | |
| 104 | 88 | 92 | 78 | 66 | 69 | 26 | 22 | 23 | |
| $S_1=\{P_1,P_5,P_9,P_{13}\}$ | | | $S_2=\{P_2,P_6,P_{10},P_{14}\}$ | | | $S_3=\{P_3,P_7,P_{11},P_{15}\}$ | | | $S_4=\{P_4,P_8,P_{12},P_{16}\}$ |

**Figure 1.** The matrix $\mathcal{A} \otimes \mathcal{B}_{21}$ is the Krönecker product of $\mathcal{A}$ and $\mathcal{B}_{21}$ as in Equation (7), and is a secret sharing scheme with reconstruction number 2. A secret reconstruction algorithm for this scheme is detailed in Section 5.1.

*5.1. Secret Reconstruction*

The matrix $\mathcal{A} \otimes \mathcal{B}_{21}$ in the above example produces interesting results.

1. A collection of three players—exactly two from one of the sets $T_1, T_2.T_3, T_4$ and one from another—allows reconstruction of the secret. For example, consider the set of three players $\{P_1, P_2, P_5\}$. This set can reconstruct the secret:

    (i) $\gcd(22, 23, 24, 23, 24, 26) = 1$; hence, the first row of $M_{\mathcal{A}}$ is (1 2 3) and the first two rows of $M_{\mathcal{B}}$ are (22 23 24) and (23 24 26). As $\tau_2 = 2$, $M_{\mathcal{B}}$ can be obtained from its two rows.

    (ii) Now, observing $5 = 4 \cdot 1 + 1$, we readily know $P_5$ uses the first row of $M_{\mathcal{B}}$ and the second row of $M_{\mathcal{A}}$; this yields the second row of $M_{\mathcal{A}}$, (2 1 4). Since $\tau_1 = 2$ and we have two rows of $M_{\mathcal{A}}$, the whole matrix $M_{\mathcal{A}}$ is known.

2. Any collection of three players—two from one of the sets $S_1, S_2, S_3, S_4$ and one from another—also allows reconstruction of the secret.

3. Reconstruction of the secret is ensured for a collection of five or more players.

    This idea can be generalized to a secret reconstruction algorithm in the general case.

## 6. Share Repair for a Krönecker Product-Induced Distribution Design

Let $\mathcal{A}$ and $\mathcal{B}$ be $(v_1, k_1, 1)$- and $(v_2, k_2, 1)$-BIBDs with $b_1$ and $b_2$ blocks, and replication numbers $r_1$ and $r_2$, respectively. Consider player $P_1$, whose share is the first block (i.e., first row) of $\mathcal{A} \otimes \mathcal{B}$. Thus,

$$\text{share of } P_1 = \mathfrak{a}_{11}\mathfrak{b}_{11} \; \mathfrak{a}_{11}\mathfrak{b}_{12} \; \cdots \; \mathfrak{a}_{11}\mathfrak{b}_{1k_2} \mid \mathfrak{a}_{12}\mathfrak{b}_{11} \; \mathfrak{a}_{12}\mathfrak{b}_{12} \; \cdots \; \mathfrak{a}_{12}\mathfrak{b}_{1k_2} \mid \cdots$$
$$\cdots \mid \mathfrak{a}_{1k_1}\mathfrak{b}_{11} \; \mathfrak{a}_{1k_1}\mathfrak{b}_{12} \; \cdots \; \mathfrak{a}_{1k_1}\mathfrak{b}_{1k_2} = L_1 \mid L_2 \mid \cdots \mid L_{k_1}.$$

Using the notations and method described in [8] (and making the same assumption that any player is available with a fixed probability $p$), the probability of availability of at least one repair set is

$$R(p) = \left(1 - (1 - p)^{r_1 r_2}\right)^{k_1 k_2}. \tag{8}$$

We improve this method significantly. For this, observe that each block $L_k$

$(k \in \{1, 2, \ldots, k_1\})$ (possibly with a different factor $\mathfrak{a}_{mi}$ for some $m \in \{1, 2, \ldots$

$\ldots, b_1\}, i \in \{1, 2, \ldots, k_1\}$, from $\mathcal{A}$) occurs in the shares of $r_1 - 1$ players other than $P_1$.    (9)

Furthermore, the share of $P_1$ can also be characterized as

$$\mathfrak{a}_{11}\mathfrak{b}_{11} \, \mathfrak{a}_{11}\mathfrak{b}_{12} \, \cdots \, \mathfrak{a}_{11}\mathfrak{b}_{1k_2} \mid \mathfrak{a}_{12}\mathfrak{b}_{11} \, \cdots \, \mathfrak{a}_{12}\mathfrak{b}_{1k_2} \mid \cdots\cdots \mid \mathfrak{a}_{1k_1}\mathfrak{b}_{11} \, \cdots \, \mathfrak{a}_{1k_1}\mathfrak{b}_{1k_2};$$

$$
\begin{aligned}
K_1 &:= \mathfrak{a}_{11}\mathfrak{b}_{11} \, \mathfrak{a}_{12}\mathfrak{b}_{11} \, \cdots \, \mathfrak{a}_{1k_1}\mathfrak{b}_{11}, \\
K_2 &:= \mathfrak{a}_{11}\mathfrak{b}_{12} \, \mathfrak{a}_{12}\mathfrak{b}_{12} \, \cdots \, \mathfrak{a}_{1k_1}\mathfrak{b}_{12}, \\
&\;\;\vdots \\
K_{k_2} &:= \mathfrak{a}_{11}\mathfrak{b}_{1k_2} \, \mathfrak{a}_{12}\mathfrak{b}_{1k_2} \, \cdots \, \mathfrak{a}_{1k_1}\mathfrak{b}_{1k_2}.
\end{aligned}
$$

It is thus clear that each $K_j (j \in \{1, 2, \ldots, k_2\})$ (possibly with a different

factor $\mathfrak{b}_{lj}$ for some $l \in \{1, 2, \ldots, b_2\}$, from $\mathcal{B}$) occurs in the shares of $r_2 - 1$

players other than $P_1$.    (10)

Let us assume that we have $t_1$ players of type (9) and $t_2$ players of type (10). Then

$$R^*_{(t_1, t_2)}(p) = R^*_{t_1}(p) R^*_{t_2}(p) R^*_{\delta}(p), \tag{11}$$

where

(i)      $t_1$ are selected from type (9);
(ii)     $t_2$ are selected from type (10);
(iii)    $\delta := k_1 k_2 - t_1 k_1 - t_2 (k_2 - t_1)$ are selected independently, and

$$
\begin{aligned}
R^*_{t_1}(p) &= \left(1 - (1-p)^{r_1 - 1}\right)^{t_1} \\
R^*_{t_2}(p) &= \left(1 - (1-p)^{r_2 - 1}\right)^{t_2} \\
R^*_{\delta}(p) &= \left(1 - (1-p)^{(r_1 - 1)(r_2 - 1)}\right)^{\delta}.
\end{aligned}
$$

Observe that $\delta = (k_1 - t_2)(k_2 - t_1)$. Therefore, the probability of at least one repair set being available in this case is

$$R^*(p) = \sum_{t_1, t_2} R^*_{t_1}(p) R^*_{t_2}(p) R^*_{\delta}(p).$$

Let $E^*(p)$ be the expected number of minimal repair sets. In general, this expected number is the product of the total number of possible repair sets and the probability of availability of each repair set. Ref. [8] sets $E(p) = (r_1 r_2)^{k_1 k_2}$. We denote by $C(t_1, t_2)$, the

number of partitions of a set of size $k_1 k_2$ into three sets of sizes $t_1$, $t_2$ and $k_1 k_2 - t_1 - t_2$. By an argument similar to the previous,

$$
\begin{aligned}
E_{t_1}^*(p) &= (r_1 - 1)^{t_1} p^{t_1}, \\
E_{t_2}^*(p) &= (r_2 - 1)^{t_2} p^{t_2}, \text{ and} \\
E_\delta^*(p) &= [(r_1 - 1)(r_2 - 1)]^\delta p^\delta, \text{ so that} \\
E_{(t_1, t_2)}^*(p) &= C(t_1, t_2) E_{t_1}^*(p) E_{t_2}^*(p) E_\delta^*(p). \\
\text{Hence, } E^*(p) &= \sum_{t_1, t_2} C(t_1, t_2) E_{t_1}^*(p) E_{t_2}^*(p) E_\delta^*(p).
\end{aligned}
$$

Table 1 shows a comparison of share repair probability on three projective plains for two different methods.

**Table 1.** A comparison table showing probability of share repairability on three projective planes.

| $\mathcal{A}$ | $\mathcal{B}$ | $R(p)$ | $R^*(p)$ |
|:---:|:---:|:---:|:---:|
| $(3, 2, 1)$ | $(3, 2, 1)$ | $(1 - q^3)^4$ | $> (1 - q)^4 + \ldots$ |
| $(3, 2, 1)$ | $(7, 3, 1)$ | $(1 - q^5)^6$ | $> (1 - q^2)^6 + \ldots$ |
| $(7, 3, 1)$ | $(7, 3, 1)$ | $(1 - q^8)^9$ | $> (1 - q^4)^9 + \ldots$ |

## 7. Frameproofness

Consider matrix representations of two BIBDs $\mathcal{A} = (\mathfrak{a}_{ij})_{\substack{i \in \{1, \ldots, b_1\} \\ j \in \{1, \ldots, k_1\}}}$ and $\mathcal{B} = (\mathfrak{b}_{ij})_{\substack{i \in \{1, \ldots, b_2\} \\ j \in \{1, \ldots, k_2\}}}$, and their Krönecker product as depicted in Equation (4). We show here how the share of a player, say $P_1$, can be retrieved (i.e., player $P_1$ can be *framed*; see [10] for more details) by only two other players. For clarity, we mention here that the share of $P_1$ is $\mathfrak{a}_{11}\mathfrak{b}_{11}$, $\mathfrak{a}_{11}\mathfrak{b}_{12}$, $\ldots$, $\mathfrak{a}_{12}\mathfrak{b}_{11}$, $\mathfrak{a}_{12}\mathfrak{b}_{12}$, $\ldots$, $\mathfrak{a}_{13}\mathfrak{b}_{11}$, $\ldots$.

1. There exist $(b_2 - 1) + (r_1 - 1) \cdot b_2$ players that possess the element $\mathfrak{a}_{11}\mathfrak{b}_{ij}$ for some $i \in \{1, 2, \ldots, b_2\}$ and $j \in \{1, 2, \ldots, k_2\}$, since $r_1$ is the replication number of $\mathcal{A}$. Of these, $(r_1 - 1) \cdot 1$ players possess the first $k_2$ elements of the share, i.e., $\mathfrak{a}_{11}\mathfrak{b}_{11}$ $\mathfrak{a}_{11}\mathfrak{b}_{12}$ $\ldots$ $\mathfrak{a}_{11}\mathfrak{b}_{1k_2}$. If any of these players know the ratios $\frac{\mathfrak{a}_{12}}{\mathfrak{a}_{11}}$, $\frac{\mathfrak{a}_{13}}{\mathfrak{a}_{11}}$, $\ldots$, then they could construct the entire share of $P_1$.

2. Note that for $j \neq 1$, any of the $b_2 - 1$ players with shares

$$
\begin{aligned}
\mathfrak{a}_{11}\mathcal{B}_2 \mid \mathfrak{a}_{12}\mathcal{B}_2 \mid \quad &\ldots \quad \mid \mathfrak{a}_{1k_1}\mathcal{B}_2, \\
\mathfrak{a}_{11}\mathcal{B}_3 \mid \mathfrak{a}_{12}\mathcal{B}_3 \mid \quad &\ldots \quad \mid \mathfrak{a}_{1k_1}\mathcal{B}_3, \\
&\vdots \\
\mathfrak{a}_{11}\mathcal{B}_{b_2} \mid \mathfrak{a}_{12}\mathcal{B}_{b_2} \mid \quad &\ldots \quad \mid \mathfrak{a}_{1k_1}\mathcal{B}_{b_2}
\end{aligned}
$$

know these ratios.

Therefore, only two players — one from the $r_1 - 1$ players possessing $\mathfrak{a}_{11}\mathfrak{b}_{11}$ and one from the $b_2 - 1$ players possessing $\frac{\mathfrak{a}_{12}}{\mathfrak{a}_{11}}$, $\frac{\mathfrak{a}_{13}}{\mathfrak{a}_{11}}$, $\ldots$ — can reconstruct the entire share of player $P_1$, and hence, frame this player.

We try to address this problem by reducing the repetitive nature of shares of the participants. We shall do this by decreasing the size of each share, while retaining all the information that a player had in the previous construction (i.e., Equation (4)).

### 7.1. A Modified Scheme

Given two matrices $\mathcal{A}$ and $\mathcal{B}$ of the same dimension $r \times c$, we define the operation $\mathcal{A} \odot \mathcal{B}$ as the $r \times c$ matrix generated by position-wise products of elements of $\mathcal{A}$ and $\mathcal{B}$, i.e.,

$$\text{if } \mathcal{A} = \begin{pmatrix} \mathfrak{a}_{11} & \mathfrak{a}_{12} & \cdots & \mathfrak{a}_{1c} \\ \vdots & & & \\ \mathfrak{a}_{r1} & \mathfrak{a}_{r2} & \cdots & \mathfrak{a}_{rc} \end{pmatrix} \text{ and } \mathcal{B} = \begin{pmatrix} \mathfrak{b}_{11} & \mathfrak{b}_{12} & \cdots & \mathfrak{b}_{1c} \\ \vdots & & & \\ \mathfrak{b}_{r1} & \mathfrak{b}_{r2} & \cdots & \mathfrak{b}_{rc} \end{pmatrix}, \text{ then}$$

$$\mathcal{A} \odot \mathcal{B} = \begin{pmatrix} \mathfrak{a}_{11}\mathfrak{b}_{11} & \mathfrak{a}_{12}\mathfrak{b}_{12} & \cdots & \mathfrak{a}_{1c}\mathfrak{b}_{1c} \\ \vdots & & & \\ \mathfrak{a}_{r1}\mathfrak{b}_{r1} & \mathfrak{a}_{r2}\mathfrak{b}_{r2} & \cdots & \mathfrak{a}_{rc}\mathfrak{b}_{rc} \end{pmatrix}.$$

The operator $\odot$ is well-behaved in the sense that it is commutative and respects scalar multiplication on integer-valued matrices.

Let $\pi : \{1, 2, \ldots, b\} \to \{1, 2, \ldots, b\}$ be a permutation. Given $i \in \{1, 2, \ldots, b\}$ and $\pi(i) = j$, we define $\tilde{\pi} : \{1, 2, \ldots, b\} \to \{1, 2, \ldots, k\}$ as $\tilde{\pi}(i) = j \pmod{k}$, for any integer $k \leq b$. Now given BIBDs $\mathcal{A}_{b_1 \times k_1}$ and $\mathcal{B}_{b_2 \times k_2}$, we modify their Krönecker product by first choosing a permutation $\pi_1$ randomly from the set of all permutations over $\{1, 2, \ldots, b_2\}$ and producing $\tilde{\pi}_1$. Then we produce $\tilde{\pi}_2, \tilde{\pi}_3, \ldots, \tilde{\pi}_{k_1}$ by simple translations.

Next, we represent application of the function $\tilde{\pi}_l$ to the $m^{\text{th}}$ block matrix (of size $b_2 \times k_2$) of block-row $t$ in $\mathcal{A} \otimes \mathcal{B}$ by $\theta_{mt} = l$, and define matrix $N_{b_1 b_2 \times k_1 k_2} = (n_{ij})$ divided into blocks of size $b_2 \times k_2$ similarly as $\mathcal{A} \otimes \mathcal{B}$ such that

$$\begin{cases} n_{ij} = 1 & \text{if } \tilde{\pi}_l(i) = j \\ n_{ij} = 0 & \text{if otherwise} \end{cases},$$

where $n_{ij}$ is the element in the $i$th row and $j$th column of the $(m, t)$th block matrix of $M$. Finally, the $i$th row of matrix $(\mathcal{A} \otimes \mathcal{B}) \odot N$ produces the share of player $P_i$ ($i \in \{1, 2, \ldots, b_1 b_2\}$) by omitting the zeroes.

### 7.2. Example

Consider another example, where a $2 - (4, 3, 2)$-BIBD and a $2 - (5, 4, 3)$-BIBD over the points $\{1, 2, 3, 4\}$ and $\{22, 23, 24, 25, 26\}$ are represented by matrices $\mathcal{A}$ and $\mathcal{B}$, respectively (note that $r_1 = 3, r_2 = 4$):

$$\mathcal{A} = \begin{pmatrix} 1 & 2 & 3 \\ 2 & 3 & 4 \\ 3 & 4 & 1 \\ 4 & 1 & 2 \end{pmatrix}, \text{ and } M_{\mathcal{B}} = \begin{pmatrix} 22 & 23 & 24 & 25 \\ 23 & 24 & 25 & 26 \\ 24 & 25 & 26 & 22 \\ 25 & 26 & 22 & 23 \\ 26 & 22 & 23 & 24 \end{pmatrix}. \tag{12}$$

Then $b_1 = 4, b_2 = 5, k_1 = 3$ and $k_2 = 4$; $\tau_1 = 2$ and $\tau_2 = 2$ are the reconstruction numbers of $\mathcal{A}$ and $\mathcal{B}$, respectively.

Modifying the matrix in Figure 2, as shown in Figures 3 and 4, we obtain a scheme for which it is no longer possible to reconstruct the secret of the scheme in Figure 4 from just two players (as was possible in the example in Section 5). In fact, the proceeding section provides an algorithm for secret reconstruction from this scheme using $\tau_1 + \tau_2$ players.

| 22 | 23 | 24 | 25 | 44 | 46 | 48 | 50 | 66 | 69 | 72 | 75 |
| 23 | 24 | 25 | 26 | 46 | 48 | 50 | 52 | 69 | 72 | 75 | 78 |
| 24 | 25 | 26 | 22 | 48 | 50 | 52 | 44 | 72 | 75 | 78 | 66 |
| 25 | 26 | 22 | 23 | 50 | 52 | 44 | 46 | 75 | 78 | 66 | 69 |
| 26 | 22 | 23 | 24 | 52 | 44 | 46 | 48 | 78 | 66 | 69 | 72 |
| 44 | 46 | 48 | 50 | 66 | 69 | 72 | 75 | 88 | 92 | 96 | 100 |
| 46 | 48 | 50 | 52 | 69 | 72 | 75 | 78 | 92 | 96 | 100 | 104 |
| 48 | 50 | 52 | 44 | 72 | 75 | 78 | 66 | 96 | 100 | 104 | 88 |
| 50 | 52 | 44 | 46 | 75 | 78 | 66 | 69 | 100 | 104 | 88 | 92 |
| 52 | 44 | 46 | 48 | 78 | 66 | 69 | 72 | 104 | 88 | 92 | 96 |
| 66 | 69 | 72 | 75 | 88 | 92 | 96 | 100 | 22 | 23 | 24 | 25 |
| 69 | 72 | 75 | 78 | 92 | 96 | 100 | 104 | 23 | 24 | 25 | 26 |
| 72 | 75 | 78 | 66 | 96 | 100 | 104 | 88 | 24 | 25 | 26 | 22 |
| 75 | 78 | 66 | 69 | 100 | 104 | 88 | 92 | 25 | 26 | 22 | 23 |
| 78 | 66 | 69 | 72 | 104 | 88 | 92 | 96 | 26 | 22 | 23 | 24 |
| 88 | 92 | 96 | 100 | 22 | 23 | 24 | 25 | 44 | 46 | 48 | 50 |
| 92 | 96 | 100 | 104 | 23 | 24 | 25 | 26 | 46 | 48 | 50 | 52 |
| 96 | 100 | 104 | 88 | 25 | 26 | 22 | 23 | 48 | 50 | 52 | 44 |
| 100 | 104 | 88 | 92 | 25 | 26 | 22 | 23 | 50 | 52 | 44 | 46 |
| 104 | 88 | 92 | 96 | 26 | 22 | 23 | 24 | 52 | 44 | 46 | 48 |

**Figure 2.** The matrix $\mathcal{A} \otimes \mathcal{B}$ is the Krönecker product of $\mathcal{A}$ and $\mathcal{B}$ as in Equation (12), and is a secret sharing scheme with reconstruction number 2.

| 1 | 0 | 0 | 0 | 0 | 0 | 0 | 1 | 0 | 0 | 1 | 0 |
| 1 | 0 | 0 | 0 | 1 | 0 | 0 | 0 | 0 | 0 | 0 | 1 |
| 0 | 1 | 0 | 0 | 1 | 0 | 0 | 0 | 1 | 0 | 0 | 0 |
| 0 | 0 | 1 | 0 | 0 | 1 | 0 | 0 | 1 | 0 | 0 | 0 |
| 0 | 0 | 0 | 1 | 0 | 0 | 1 | 0 | 0 | 1 | 0 | 0 |
| 0 | 0 | 0 | 1 | 0 | 0 | 1 | 0 | 1 | 0 | 0 | 0 |
| 1 | 0 | 0 | 0 | 0 | 0 | 0 | 1 | 1 | 0 | 0 | 0 |
| 1 | 0 | 0 | 0 | 1 | 0 | 0 | 0 | 0 | 1 | 0 | 0 |
| 0 | 1 | 0 | 0 | 1 | 0 | 0 | 0 | 0 | 0 | 1 | 0 |
| 0 | 0 | 1 | 0 | 0 | 1 | 0 | 0 | 0 | 0 | 0 | 1 |
| 0 | 0 | 1 | 0 | 1 | 0 | 0 | 0 | 0 | 0 | 0 | 1 |
| 0 | 0 | 0 | 1 | 1 | 0 | 0 | 0 | 1 | 0 | 0 | 0 |
| 1 | 0 | 0 | 0 | 0 | 1 | 0 | 0 | 1 | 0 | 0 | 0 |
| 1 | 0 | 0 | 0 | 0 | 0 | 1 | 0 | 0 | 1 | 0 | 0 |
| 0 | 1 | 0 | 0 | 0 | 0 | 0 | 1 | 0 | 0 | 1 | 0 |
| 1 | 0 | 0 | 0 | 0 | 0 | 0 | 1 | 0 | 0 | 1 | 0 |
| 1 | 0 | 0 | 0 | 1 | 0 | 0 | 0 | 0 | 0 | 0 | 1 |
| 0 | 1 | 0 | 0 | 1 | 0 | 0 | 0 | 1 | 0 | 0 | 0 |
| 0 | 0 | 1 | 0 | 0 | 1 | 0 | 0 | 1 | 0 | 0 | 0 |
| 0 | 0 | 0 | 1 | 0 | 0 | 1 | 0 | 0 | 1 | 0 | 0 |

**Figure 3.** The matrix *N*, right-operated as $\odot N$ on the tensor design $\mathcal{A} \otimes \mathcal{B}$ in Figure 2

Left matrix $(\mathcal{A} \otimes \mathcal{B}) \odot N$:

| 22 | 0 | 0 | 0 | 0 | 0 | 0 | 50 | 0 | 0 | 72 | 0 |
|----|---|---|---|---|---|---|----|---|---|----|---|
| 23 | 0 | 0 | 0 | 46 | 0 | 0 | 0 | 0 | 0 | 0 | 78 |
| 0 | 25 | 0 | 0 | 48 | 0 | 0 | 0 | 72 | 0 | 0 | 0 |
| 0 | 0 | 22 | 0 | 0 | 52 | 0 | 0 | 75 | 0 | 0 | 0 |
| 0 | 0 | 0 | 24 | 0 | 0 | 46 | 0 | 0 | 66 | 0 | 0 |
| 0 | 0 | 0 | 50 | 0 | 0 | 72 | 0 | 88 | 0 | 0 | 0 |
| 46 | 0 | 0 | 0 | 0 | 0 | 0 | 78 | 92 | 0 | 0 | 0 |
| 48 | 0 | 0 | 0 | 72 | 0 | 0 | 0 | 0 | 100 | 0 | 0 |
| 0 | 52 | 0 | 0 | 75 | 0 | 0 | 0 | 0 | 0 | 88 | 0 |
| 0 | 0 | 46 | 0 | 0 | 66 | 0 | 0 | 0 | 0 | 0 | 96 |
| 0 | 0 | 72 | 0 | 88 | 0 | 0 | 0 | 0 | 0 | 0 | 25 |
| 0 | 0 | 0 | 78 | 92 | 0 | 0 | 0 | 23 | 0 | 0 | 0 |
| 72 | 0 | 0 | 0 | 0 | 100 | 0 | 0 | 24 | 0 | 0 | 0 |
| 75 | 0 | 0 | 0 | 0 | 104 | 0 | 0 | 0 | 26 | 0 | 0 |
| 0 | 66 | 0 | 0 | 0 | 0 | 92 | 0 | 0 | 0 | 23 | 0 |
| 88 | 0 | 0 | 0 | 0 | 0 | 0 | 25 | 0 | 0 | 48 | 0 |
| 92 | 0 | 0 | 0 | 23 | 0 | 0 | 0 | 0 | 0 | 0 | 52 |
| 0 | 100 | 0 | 0 | 25 | 0 | 0 | 0 | 48 | 0 | 0 | 0 |
| 0 | 0 | 88 | 0 | 0 | 26 | 0 | 0 | 50 | 0 | 0 | 0 |
| 0 | 0 | 0 | 96 | 0 | 0 | 23 | 0 | 0 | 44 | 0 | 0 |

$\mapsto$

Right matrix (share distribution scheme):

| 22 | 50 | 72 |
|----|----|----|
| 23 | 46 | 78 |
| 25 | 48 | 72 |
| 22 | 52 | 75 |
| 24 | 46 | 66 |
| 50 | 72 | 88 |
| 46 | 78 | 92 |
| 48 | 72 | 100 |
| 52 | 75 | 88 |
| 46 | 66 | 96 |
| 72 | 88 | 25 |
| 78 | 92 | 23 |
| 72 | 100 | 24 |
| 75 | 104 | 26 |
| 66 | 92 | 23 |
| 88 | 25 | 48 |
| 92 | 23 | 52 |
| 100 | 25 | 48 |
| 88 | 26 | 50 |
| 96 | 23 | 44 |

**Figure 4.** The matrix on the left is $(\mathcal{A} \otimes \mathcal{B}) \odot N$, and the one on the right is the share distribution scheme obtained from this operation, as described in Section 7.1.

### 7.3. Secret Reconstruction for the Modified Scheme

1.  Choose a player $P_i^m$ (which is the $i$th player in the $m$th row-block of $\mathcal{A} \otimes \mathcal{B}$, or the $((m-1)b_2 + i)$th player from the top), for any $m \in \{1, 2, \ldots, b_1\}$ and $i \in \{1, 2, \ldots, b_2\}$.
2.  Consider elements $a_{mt}b_{ij}$ in the share of player $P_i^m$, i.e., $\theta_{mt} = l$ and $\tilde{\pi}^l(i) = j$. For such an element $a_{mt}b_{ij}$, set $y = b_{ij}$ (note that the value $y \in \{y_1, y_2, \ldots, y_{v_2}\}$ is not known, but the positions at which the matrix $\mathcal{B}$ contains elements $b_{\hat{i}\hat{j}} = y$ is known).
3.  Construct set $\mathcal{S}_y := \left\{ \hat{l} : \left( \tilde{\pi}^{\hat{l}}(\hat{i}) = \hat{j} \right) \wedge \left( b_{\hat{i}\hat{j}} = y \right) \right\}$. By Theorem 6, for a maximal set $\mathcal{S}_y$ (if not, then another value $y$ may be chosen by selecting a different element $a_{m't'}b_{i'j'}$) the set

$$\left\{ a_{\hat{m}\hat{i}} \quad : \quad a_{\hat{m}\hat{i}} b_{\hat{i}\hat{j}} \in \text{ the share of player } P_{\hat{i}}^{\hat{m}} \text{ such that } b_{\hat{i}\hat{j}} = y \right\}$$
$$= \quad \{x_1, x_2, \ldots, x_{v_1}\}$$

    is the set of all values in $\mathcal{A}$.
4.  Construct matrix $\mathcal{A}$, since the positions of all values $x_1, x_2, \ldots, x_{v_1}$ in this matrix are now known.
5.  Compute $b_{i'j'}$ for $a_{m't'}b_{i'j'} \in$ share of player $P_{i'}^{m'}$ using the known values $a_{m't'}$ until all values $y_1, y_2, \ldots, y_{v_2}$ are known.
6.  Construct matrix $\mathcal{B}$, since the positions of all values $y_1, y_2, \ldots, y_{v_2}$ in this matrix are now known.
7.  Compute $\mathcal{A} \otimes \mathcal{B}$ from the two known matrices.

Thus, framing any player is not possible for just two other participants, and requires a much larger coalition.

## 8. Graphical Representation and Proof of Existence of Permutations
### Matching in Bipartite Graphs

Given an undirected graph $\mathcal{G}$, a *matching* of $\mathcal{G}$ is a subgraph $\mathcal{M}$ containing all vertices of $\mathcal{G}$ such that each vertex in $\mathcal{M}$ has either 0 or 1 edge incident to it. $\mathcal{M}$ is a *maximal matching* of $\mathcal{G}$ if it is not a subgraph of any other matching of $\mathcal{G}$. Thus, adding even one more edge to a maximal matching $\mathcal{M}$ ensures that it is no longer a matching. The number of edges in a maximal matching of $\mathcal{G}$ is called the *matching number* of $\mathcal{G}$.

A *perfect matching* $\mathcal{M}$ of $\mathcal{G}$ is such that each vertex of $\mathcal{M}$ has an edge incident to it. Furthermore, a *vertex cover* of a graph $\mathcal{G}$ is a subgraph containing all edges of $\mathcal{G}$ such that every edge is incident to at least one vertex in the subgraph, and an *edge cover* of a graph $\mathcal{G}$ is a subgraph containing all vertices of $\mathcal{G}$ such that every vertex has at least one edge incident to it. Thus, if $\mathcal{G}$ has no isolated vertices, then the sum of the number of vertices in its minimal vertex cover and the number of edges in its minimal edge cover equals the total number of its vertices.

If the vertex set $\mathcal{V}$ of a graph $\mathcal{G}$ can be partitioned into two disjoint subsets as $\mathcal{V} = \mathcal{A} \sqcup \mathcal{B}$ such that any edge from a vertex in $\mathcal{A}$ can only be incident to a vertex in $\mathcal{B}$ and vice versa, then $\mathcal{G}$ is called a *bipartite graph*. Let us recall some interesting results on matching in bipartite graphs.

**Theorem 4** (König, [12])**.** *In any bipartite graph, the number of edges in a maximum matching equals the number of vertices in a minimum vertex cover.*

**Theorem 5** (Hall, [13])**.** *Given a bipartite graph $\mathcal{G} = (\mathcal{V}, \mathcal{E})$ with $\mathcal{V} = \mathbf{A} \sqcup \mathbf{B}$, $\mathcal{G}$ has a matching of size $|\mathbf{A}|$ if and only if for every $S \subseteq \mathbf{A}$ we have $|N(S)| \geq |S|$, where $N(S) = \{b \in \mathbf{B} : \exists a \in S \text{ with } (a, b) \in \mathcal{E}\}$.*

Figure 5 shows a bipartite graph for the tensor design.

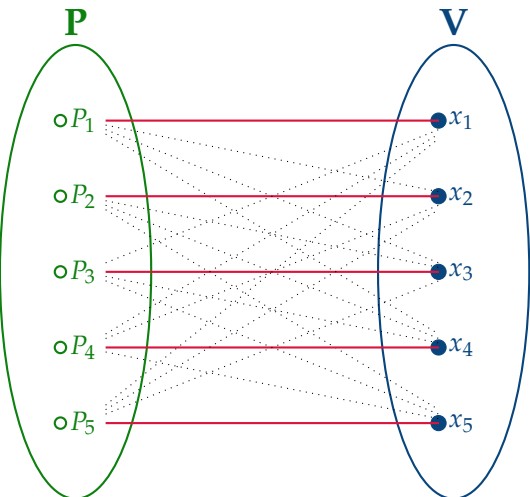

**Figure 5.** A bipartite graph for the tensor design $\mathcal{B}$ defined in Sect. with 5 players and 5 points. Each edge $(P_i, x_j)$ denotes the inclusion of point $x_j$ in the share of player $P_i$. The collection of red edges shows one possible maximal matching for the graph.

**Definition 3.** *A bipartite graph $\mathcal{G} = (\mathcal{V}, \mathcal{E})$ is said to* induce *a tensor design $\mathcal{B}$ if*

- *the vertex set $\mathcal{V} = \mathbf{P} \sqcup \mathbf{V}$ the disjoint union of the set of players $\mathbf{P} = \{P_1, \ldots, P_b\}$ and the set of points $\mathbf{V} = \{x_1, \ldots, x_v\}$ of $\mathcal{B}$;*
- *the edge set is the collection $\bigcup_{\substack{i \in [b] \\ j \in [v]}} \{(P_i, x_j) : x_j \in \text{ share of } P_i\}$.*

**Theorem 6.** *Given a bipartite graph $\mathcal{G}$ inducing a tensor design $\mathcal{B}$, and given subsets $\delta(P_i) \subseteq N(P_i)$ of size $s$,*

*(i)   If $\bigcup_{i \in [b]} \delta(P_i) = \mathbf{V}$, then reconstruction of the modified scheme $(\mathcal{A} \otimes \mathcal{B})_{\text{modified}}$ is possible.*
*(ii)  If $s \geq 1$, then (i) holds.*

**Proof.** Assuming the usual notations for a tensor design, it is clear that in $\mathcal{G}$,

$$|N(x_j)| = r \,\forall\, x_j \in \mathbf{V}$$
$$|N(P_{i_1} \cap P_{i_2})| = \lambda \tag{13}$$

From Equation (13) and the inclusion-exclusion principle,

$$|N(\{x_{i_1}, \ldots, x_{i_m}\})| \geq m(r - \lambda)$$

Since $r \geq \lambda$, Hall's theorem (Theorem 5) implies $\mathcal{G}$ has a matching of size $v$, i.e., $\bigcup_{i \in [b]} \delta(P_i) = \mathbf{V}$. Thus, (i) holds by the reconstruction algorithm in Section 7.3. Now choose $\delta(P_i)$ such that each subset contains at least one point matched with $P_i$ in this matching, so that (ii) holds. This proves the theorem. $\square$

## 9. Secret Sharing Schemes and the Internet of Things

Secret sharing schemes can be used to distribute the security key amongst numerous devices in an IoT system, ensuring that no single device has access to the entire key. They are also lightweight and require less computational power compared to other cryptographic elements. Additionally, their ability to detect and prevent attacks that attempt to modify or delete parts of the secret is particularly important in IoT applications where security is critical.

For example, in a healthcare IoT system, the security of patient data is of utmost importance. Secret sharing schemes can be used to distribute the patient data amongst multiple devices, ensuring that no single device has access to the entire data. Ref. [14] proposes an AI heuristic decision algorithm, utilizing a best-first search (BFS) approach. It effectively balances energy load and reduces communication overhead in smart healthcare technologies. The utilization of homomorphic secret sharing in IoT-based e-health applications provides various advantages in terms of privacy and security. It securely distributes secret pairs among medical nodes, ensuring the confidentiality of sensitive health data during transmission and storage within the network. This is achieved by encrypting data through homomorphic secret sharing, thereby preventing unauthorized access to medical data. Access to medical records is limited to authorized entities possessing the necessary secret keys to decrypt and utilize the shared data. Thus, the incorporation of homomorphic secret sharing adds an extra layer of protection against unauthorized modifications or alterations to medical records. A generalization of this scheme to multiple levels—possibly to combine data between different hospitals or chains of healthcare providers, different states within a country, or even different countries—can be easily achieved through the Krönecker product of the individual schemes used by each hospital system. The fields on which these schemes are based provide a perfect foundation for the homomorphism, which can be easily maintained by the integer ring over which the Krönecker product is then defined.

A frameproof tensor product of multiple distribution designs can be distinctly useful for lightweight IoT applications, as it allows for a multi-level or multi-system secret sharing scheme IoT implementation in a secure and efficient manner, while detecting and preventing any attempt to modify or delete parts of the secret data. This approach ensures that even if some levels are compromised, the overall security of the system(s) remains intact.

The wide range of applicability of our generalizations can be further seen in, say, the management of massive data, such as [15], which proposes a non-interactive approach for IoT data aggregation that utilizes additive secret sharing, addressing numerous challenges including privacy concerns, security risks, high communication overhead, and user interaction. The additive secret sharing effectively masks the original data, preventing malicious analysis by the servers. The scheme also supports offline mobile users, maintains privacy, and provides efficient algorithms for result verification. However, ref. [15] only splits the secret between two servers at a time. A frameproof tensor product can be smoothly applied in this context for connecting a large number of such systems, due to the underlying

fields over which the secrets are split between servers in individual systems, as well as the generalized integer ring over which the tensor product is then defined.

Figure 6 shows an application of tensor design in multi-system IoT.

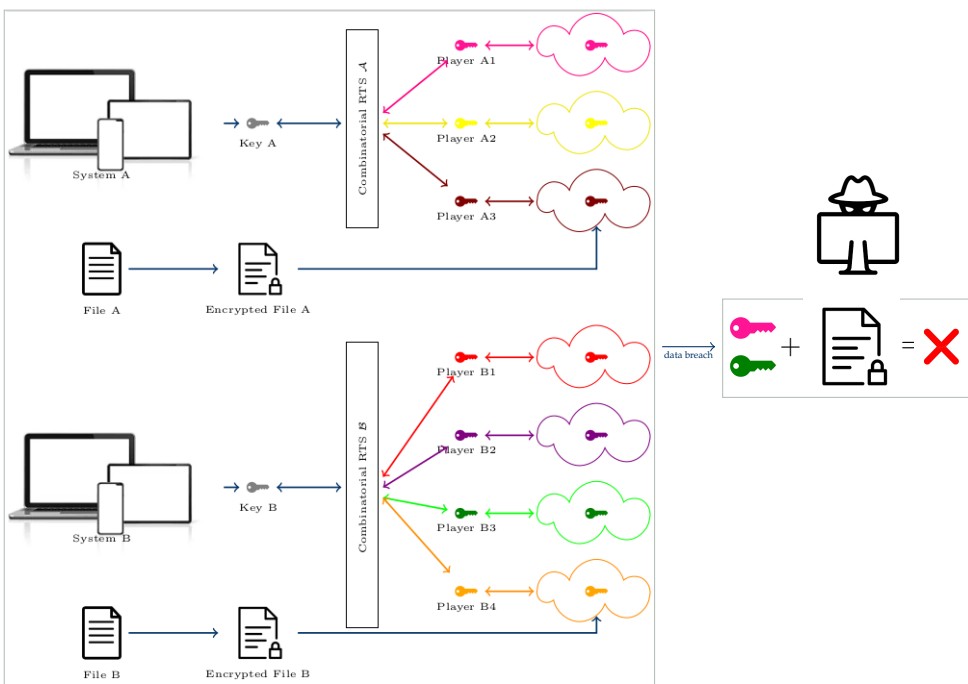

**Figure 6.** An application of the tensor product of repairable threshold schemes in multi-system IoT, where each system (say, a single hospital) may possess a separate RTS for sharing its own secret key, while multiple systems (say, a chain of hospitals) may share their individual secrets to non-colluding cloud storage providers through a tensor product of the individual schemes.

## 10. Conclusions and Future Work

In this paper, we have first generalized the concept of combinatorial RTS and then improved our secret sharing scheme by producing a frameproof one. We believe our results can be extended further to an arbitrary number of distribution designs. We also believe that the Krönecker product of BIBDs can be generalized to *t*-designs, and all corresponding results will hold for these. Furthermore, a frameproof modification for the generalized scheme also remains an open problem.

Furthermore, we have discussed the extensive scope of applicability for our proposed scheme in a diverse array of IoT contexts. A fascinating avenue for further investigation entails the examination of specific instances of these applications.

**Author Contributions:** Author B.K.R. was instrumental in providing the conceptualization of the problem. Author A.R. primarily focused on the methodology, technical details, and formal analysis. B.K.R. supplied the necessary resources for the research. A.R. contributed to the preparation of the original draft. B.K.R. played a crucial role in the review and editing of the manuscript. The supervision was conducted by B.K.R. All authors have read and agreed to the published version of the manuscript.

**Funding:** This research received no external funding.

**Data Availability Statement:** Not applicable.

**Acknowledgments:** We acknowledge Suprita Talnikar for her efforts in providing valuable insights and offering suggestions for improving the write-up. We would also like to thank the anonymous reviewers for their comments and suggestions.

**Conflicts of Interest:** The authors declare no conflict of interest.

**Sample Availability:** Not applicable.

**Abbreviations**

The following abbreviations are used in this manuscript:

| | |
|---|---|
| MDPI | Multidisciplinary Digital Publishing Institute |
| RTS | Repairable Threshold Scheme |
| BIBD | Balanced Incomplete Block Design |
| IoT | Internet of Things |
| GDPR | General Data Protection Regulation |
| SBIoT | Secret sharing-Based IoT |
| eID | electronic IDentification |
| IDA | Information Dispersal Algorithm |
| BFS | Best-First Search |

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
