# Peer review of "IoT-Applicable Generalized Frameproof Combinatorial Designs"

_2624-831X, doi:10.3390/iot4030020_

Round 1
Reviewer 1 Report
Two major comments to improve the draft:
[C1]
>Use of a ramp-type scheme improves the security and energy efficiency in SBIoT networks [2].
Give a definition of SBIoT network and explanation on what is energy efficiency.
May we ignore Communication complexity of SBIoT ?
[C2]
The authors claim:
>The scheme proposed in this paper produces a far more efficient share repairability, which 99 is >possible due to the generalized domain, and based heavily on the easier secret reconstruction >mentioned beforehand.
To support the authors’ claim above:
The revision shall give more exact comparision of the proposal to the existing schemes w.r.t. efficient by using some comparision table.//
Author Response
We thank the reviewer for the detailed review and helpful comments. We have provided the definition of SBIoT networks and an explanation of energy efficiency, which can be found on lines 35-42 on pages 1 and 2 in the revised copy.
The only existing combinatorial design proposed before this work was by Kacsmar and Stinson [8]. Hence we have included a comparison table (Table 1 on page 13) exhibiting a probability of share repair better than predicted by [8].
Furthermore, this work concentrates heavily on improving security and efficiency, for example through the concept of frameproofness. The exploration of commumication complexity of the proposed design merits thorough and separate research. We would like to express our gratitude to the reviewer for identifying this potential future work. However, we believe the proposed design does not cause significant improvement in communication complexity.
Reviewer 2 Report
The paper explores the concept of secret sharing schemes, especially focusing on repairable ramp schemes, for IoT applications. The authors propose a graph-theoretic approach to enhance these schemes by improving their share repairability and secret reconstruction while offering frameproofness. They discuss the potential applicability of their proposal in diverse IoT contexts, such as cloud storage, sensor-based IoT networks, and electronic identification cards. While they believe their scheme can be further generalized to a wider array of distribution designs, they acknowledge that more research is needed to prove this conclusively.
Strengths:
- The paper explores a novel approach to secret sharing schemes by integrating graph theory, which could potentially pave the way for more efficient and secure data sharing mechanisms.
- The paper successfully extends the concept of combinatorial Repairable Threshold Schemes (RTS) and introduces frameproofness into the scheme, improving its security aspect.
Weaknesses:
- The paper indicates the potential for further generalization of their scheme to an arbitrary number of distribution designs but it does not provide concrete proof or extensive examples to support this claim.
- While the paper is largely theoretical, including some empirical evidence or case studies could enhance its validity. If possible, conducting experiments to test the proposed schemes under various conditions and presenting the results would add substantial value.
- The claim of further generalization to an arbitrary number of distribution designs should be supported by more concrete evidence, discussions, or preliminary investigations. This could lend more credibility to the authors' conjecture.
There are a few minor instances of inappropriate wording that could be polished for clarity and conciseness. Overall, with minor improvements, the English language quality of the paper would be excellent.
Author Response
We thank the reviewer for the detailed review and helpful comments. Our main objective in this paper was to propose a construction that extends RTS. As noted, the generalisation can be extended further to an arbitrary number of distribution designs. Since it was only our intention to introduce the generalised tensor design, we did not include a detailed proof. However, our ongoing project concentrates on the generalised design as well as analyses some very interesting applications; a detailed proof shall be found in this work.
Furthermore, meticulous experimentation will be the scope of this future work.
Also we have tried our best to improve the language quality of the paper as suggested. We sincerely hope that the reviewer finds our work of acceptable standard.
Round 2
Reviewer 1 Report
Explore further research on this direction more.//
Reviewer 2 Report
The authors have successfully addressed my comments.
The English language quality of the paper is fine.